Allele specific expression and methylation in the bumblebee, Bombus terrestris

Lonsdale Zoë 1
Lee Kate 2
Kiriakidu Maria 1
Amarasinghe Harindra 3
Nathanael Despina 1
O’Connor Catherine J. 4
Mallon Eamonn B. ebm3@le.ac.uk 1
1 Department of Genetics and Genome Biology, University of Leicester , Leicester , United Kingdom
2 Bioinformatics and Biostatistics Support Hub (B/BASH), University of Leicester , Leicester , United Kingdom
3 Academic Unit of Cancer Sciences, University of Southampton , Southampton , United Kingdom
4 School of Biosciences, Cardiff University , Cardiff , United Kingdom
Berghout Joanne
Electronic publication date: 2017 Sep 13
Publication date: 2017
Volume: 5
Electronic Location ID: e3798
Received 2017 May 23; Accepted 2017 Aug 21
Copyright: ©2017 Lonsdale et al.
Copyright year: 2017
Copyright holder: Lonsdale et al.
License: This is an open access article distributed under the terms of the Creative Commons Attribution License, which permits unrestricted use, distribution, reproduction and adaptation in any medium and for any purpose provided that it is properly attributed. For attribution, the original author(s), title, publication source (PeerJ) and either DOI or URL of the article must be cited.
License URL: https://creativecommons.org/licenses/by/4.0/

Keywords: Methylation, Hymenoptera, Genomic imprinting, Allele specific expression

Funding: NERC NE/H010408/1 NE/N010019/1 R8/H10/56 NERC Biomolecular Analysis Facility research NBAF 606 and 829 MRC MR/K001744/1 BBSRC BB/J004243/1 MIBTP This work was financially supported by NERC grant no. NE/H010408/1 and NE/N010019/1 and NERC Biomolecular Analysis Facility research grants (NBAF 606 and 829) to EBM. Illumina library preparation, sequencing and bioinformatics were carried out by Edinburgh Genomics, The University of Edinburgh. Edinburgh Genomics is partly supported through core grants from NERC (R8/H10/56), MRC (MR/K001744/1) and BBSRC (BB/J004243/1). ZNL was financially supported via MIBTP. There was no additional external funding received for this study. The funders had no role in study design, data collection and analysis, decision to publish, or preparation of the manuscript.

==============================
The social hymenoptera are emerging as models for epigenetics. DNA methylation, the addition of a methyl group, is a common epigenetic marker. In mammals and flowering plants methylation affects allele specific expression. There is contradictory evidence for the role of methylation on allele specific expression in social insects. The aim of this paper is to investigate allele specific expression and monoallelic methylation in the bumblebee, Bombus terrestris. We found nineteen genes that were both monoallelically methylated and monoallelically expressed in a single bee. Fourteen of these genes express the hypermethylated allele, while the other five express the hypomethylated allele. We also searched for allele specific expression in twenty-nine published RNA-seq libraries. We found 555 loci with allele-specific expression. We discuss our results with reference to the functional role of methylation in gene expression in insects and in the as yet unquantified role of genetic cis effects in insect allele specific methylation and expression.

Introduction

Epigenetics is the study of heritable changes in gene expression that do not involve changes to the underlying DNA sequence (Goldberg, Allis & Bernstein, 2007). Social hymenoptera (ants, bees, and wasps) are important emerging models for epigenetics (Glastad et al., 2011; Weiner & Toth, 2012; Welch & Lister, 2014; Yan et al., 2014). This is due to theoretical predictions for a role for an epigenetic phenomenon, genomic imprinting (parent of origin allele specific expression), in their social organisation (Queller, 2003), the recent discovery of parent-of-origin allele specific expression in honeybees (Galbraith et al., 2016), and data showing a fundamental role in social insect biology for DNA methylation, an epigenetic marker (Chittka, Wurm & Chittka, 2012).

In mammals and flowering plants, allele specific expression is often associated with methylation marks passed from parents to offspring (Reik & Walter, 2001). However DNA methylation is involved in numerous other cellular processes (Bird, 2002). There is contradictory evidence for the role of methylation on allele specific expression in social insects. Methylation is associated with allele specific expression in a number of loci in the ants Camponotus floridanus and Harpegnathos saltator (Bonasio et al., 2012). Recently, we found evidence for allele specific expression in bumblebee worker reproduction genes (Amarasinghe et al., 2015) and that methylation is important in bumblebee worker reproduction (Amarasinghe, Clayton & Mallon, 2014). However, other work on the honeybee Apis mellifera found no link between genes showing allele specific expression and known methylation sites in that species (Kocher et al., 2015).

The presence of allele specific expression does not necessarily mean an epigenetic process is involved. Allele specific expression is known to be caused by a number of genetic as well as epigenetic processes (Palacios et al., 2009). The genetic process usually involves cis effects such as transcription factor binding sites, or less often, untranslated regions which alter RNA stability or microRNA binding (Farh et al., 2005).

The aim of this paper is to investigate allele specific expression and methylation in the bumblebee, Bombus terrestris. The recently sequenced genome of the bumblebee, Bombus terrestris displays a full complement of genes involved in the methylation system (Sadd et al., 2015). An extreme form of allele specific expression involves monoallelic expression, where one allele is completely silenced. In the canonical mammal and flowering plant systems, this is often associated with monoallelic methylation. In this paper, we examined the link between monoallelic methylation and monoallelic expression in the bumblebee, Bombus terrestris using an integrative approached previously used in human epigenetic studies (Harris et al., 2010). Namely, we compare two types of whole methylome libraries and an RNA-seq library from the same individual. In humans, this integrative approach has been independently validated by clonal bisulphite sequencing (Harris et al., 2010). MeDIP-seq is an immunoprecipitation technique that creates libraries enriched for methylated cytosines (Harris et al., 2010). Methyl-sensitive restriction enzymes can create libraries that are enriched for non-methylated cytosines (MRE-seq) (Harris et al., 2010). Genes found in both libraries are predicted to be monoallelically methylated, with the putatively hypermethylated allele being in the MeDIP-seq data and the putatively hypomethylated allele in the MRE-seq data (Harris et al., 2010). Monoallelic expression was identified in these loci from the RNA-seq library. If only one allele was expressed then we knew that these loci were both monoallelically methylated and monoallelically expressed in this bee. We confirmed this monoallelic expression in one locus using qPCR.

We then more generally searched for allele specific expression by analysing twenty nine published RNA-seq libraries from worker bumblebees (Harrison, Hammond & Mallon, 2015; Riddell et al., 2014). We identified heterozygotes in the RNA-seq libraries and measured the expression of each allele. We then identified loci that showed significant expression differences between their two alleles.

Materials and Methods

Samples

Data from twenty-nine RNA-seq libraries were used for the allele specific expression analysis (six from Harrison, Hammond & Mallon (2015), and twenty-three from Riddell et al. (2014)). The Riddell bees came from two colonies, one commercially reared bumblebee colony from Koppert Biological Systems UK and one colony from a wild caught queen from the botanic gardens, Leicester. The Harrison bees were from four commercially reared colonies obtained from Agralan Ltd. A Koppert colony worker bee was used for the MeDIP-seq / MRE-seq / RNA-seq experiment. Bees from three different Koppert colonies were used for the qPCR analysis. Samples are outlined in Table 1. Colonies were fed ad libitum with pollen (Percie du sert, France) and 50% diluted glucose/fructose mix (Meliose—Roquette, France). Before and during the experiments colonies were kept at 26 °C and 60% humidity in constant red light.

Table 1 Bees used in each experiment.

K refers to Koppert, A to Agralan and Q to the wild caught Leicester queen.

Experiment	Number	Colony	Tissue	
Allele specific expression RNA-seq	1	A1	Whole body	
	2	A2	Whole body	
	2	A3	Whole body	
	1	A4	Whole body	
	14	K1	Abdomen	
	9	Q1	Abdomen	
MeDip/MRE/RNA-seq	1	K2	Whole body	
qPCR	2	K3	Head	
	1	K4	Head	
	1	K5	Head	

Next generation sequencing

MeDIP-seq, MRE-seq and RNA-seq

RNA and DNA was extracted from a single five day old whole bee (Colony K2). DNA was extracted using an ethanol precipitation method. Total RNA was extracted using Tri-reagent (Sigma-Aldrich, UK).

Three libraries were prepared from this bee by Eurofins genomics. These were MeDIP-seq and MRE-seq libraries on the DNA sample and one amplified short insert cDNA library with size of 150–400 bp on the RNA sample. Both the MeDIP-seq and MRE-seq library preparations are based on previously published protocols (Harris et al., 2010). MeDIP-seq uses monoclonal antibodies against 5-methylcytosine to enrich for methylated DNA independent of DNA sequence. MRE-seq enriches for unmethylated cytosines by using methylation-sensitive enzymes that cut only restriction sites with unmethylated CpGs. Each library was individually indexed. Sequencing was performed on an Illumina HiSeq®2000 instrument (Illumina, Inc., San Diego, CA, USA) by the manufacturer’s protocol. Multiplexed 100 base paired-read runs were carried out yielding 9,390 Mbp for the MeDIP-seq library, 11,597 Mbp for the MRE-seq library and 8,638 Mbp for the RNA-seq library.

Previously published RNA-seq

Full details of the RNA-seq protocols used have been published previously (Harrison, Hammond & Mallon, 2015; Riddell et al., 2014). Briefly, for the Riddell bees, total RNA was extracted from twenty three individual homogenised abdomens using Tri-reagent (Sigma-Aldrich, Irvine, UK). TruSeq RNA-seq libraries were made from the 23 samples at NBAF Edinburgh. Multiplexed 50 base single-read runs was performed on an Illumina HiSeq2000 instrument (Illumina, Inc.) by the manufacturer’s protocol. For the Harrison bees, total RNA was extracted from whole bodies using a GenElute Mammalian Total RNA Miniprep kit (Sigma-Aldrich) following the manufacturers’ protocol. The six libraries were sequenced as multiplexed 50 base single-read runs on an Illumina HiSeq 2500 system in rapid mode at the Edinburgh Genomics facility of the University of Edinburgh.

Monoallelic methylation and expression—bioinformatic analysis

We searched for genes that were monoallelically methylated (present in both MeDip-seq (the putatively hypermethylated allele) and MRE-seq (the putatively hypomethylated allele) libraries), heterozygous (different alleles in the methylation libraries) and monoallelically expressed (only one allele present in the RNA-seq library).

Alignment and bam refinement

mRNA reads were aligned to the Bombus terrestris genome assembly (AELG00000000) using Tophat (Kim et al., 2013) and converted to bam files with Samtools (Li et al., 2009). Reads were labelled with the AddOrReplaceReadGroups.jar utility in Picard (http://picard.sourceforge.net/). The MRE-seq and MeDIP-seq reads were aligned to the genome using BWA mapper (Li & Durbin, 2009). The resultant sam alignments were soft-clipped with the CleanSam.jar utility in Picard and converted to bam format with Samtools. The Picard utility AddOrReplaceReadGroups.jar was used to label the MRE and MeDIP reads which were then locally re-aligned with GATK (DePristo et al., 2011; McKenna et al., 2010). PCR duplicates for all bams (mRNA, MeDIP and MRE) were marked with the Picard utility Markduplicates.jar.

Identifying regions of interest and integrating data

Coverage of each data type was calculated using GATK DepthofCoverage (McKenna et al., 2010). Only regions with a read depth of at least six in each of the libraries (RNA-seq, MeDIP-seq and MRE-seq) was used. Heterozygotes were identified using Samtools mpileup and bcftools on each data set separately (Li & Durbin, 2009) and results were merged with vcf tools (Danecek et al., 2011). Regions of mRNA with overlaps of MeDIP, MRE, and monoallelic snps were identified with custom perl scripts.

Allele specific expression—bioinformatic analysis

We created a pipeline to search for heterozygous loci that show allele specific expression and identify the associated enriched gene ontology (GO) terms in twenty-nine previously published RNA-seq libraries (Harrison, Hammond & Mallon, 2015; Riddell et al., 2014).

Each RNA library was mapped to the Bombus terrestris reference genome (Bter 1.0, accession AELG00000000.1) (Sadd et al., 2015) using the BWA mapper (Li & Durbin, 2009). The combat method in the R package SVA (version 3.20.0) was used to remove any batch effects and control for original differences in coverage (Leek et al., 2012; Johnson, Li & Rabinovic, 2007). The success of this control was confirmed by the R package edgeR (version 3.14.0) (McCarthy, Chen & Smyth, 2012; Robinson, McCarthy & Smyth, 2010).

Bcftools (version 0.1.19-44428cd), bedtools (version 2.17.0), and SAMtools (version 0.1.19-44428cd) were used to prepare the RNA libraries and call the SNPs, before the SNPs were filtered based on mapping quality score (Quinlan & Hall, 2010; Li & Durbin, 2009). Only SNPs with a mapping quality score of p < 0.05 and a read depth of ≥6 were included in the analyses.

The R package, QuASAR implements a statistical method for: (1) genotyping from next-generation sequencing reads (according to the Hardy–Weinberg equilibrium), and (2) conducting inference on allele specific expression at heterozygous sites (Harvey et al., 2015). One problem with genotyping heterozygotes is being able to identical true homozygotes that appear heterozygote due to base-calling errors. QuASAR removes snps with extreme differential allele expression from the analyses, thus controlling for any base-calling errors. Despite this inherent conservatism, in benchmark tests, QuaSAR can accurately genotype loci with lower error rates than other methods commonly used for genotyping DNA-seq data (Harvey et al., 2015). The allele specific expression inference step takes into consideration the uncertainty in the genotype calls, base-call errors in sequencing, and allelic over-dispersion. QuASAR is a powerful tool for detecting allele specific expression if, as during most RNA-seq experiments, genotypes are not available (Harvey et al., 2015).

Sequence regions (the snp position +∕ − 2,900 bp), encompassing the loci identified as showing ASE in at least three of the thirty libraries, were compared to Drosophila melanogaster proteins (non-redundant (nr) database) with Blastx (Altschul et al., 1997). The blast results were annotated using Blast2Go (Gotz et al., 2008). We carried out an enrichment analysis (Fisher exact test) using a custom R script (https://dx.doi.org/10.6084/m9.figshare.3201355.v1) on this list of GO terms. This identified GO terms that are overrepresented (p < 0.05) relative to the entire bumblebee transcriptome (https://dx.doi.org/10.6084/m9.figshare.3458828.v1). We then used REVIGO to summarize and visualise these terms (Supek et al., 2011). REVIGO summarizes lists of GO terms using a clustering algorithm based on semantic similarity measures. To identify which bumblebee genes the snps were located in, the snp position +∕ − 25 bp was compared against the Bombus terrestris genome (Sadd et al., 2015) using Blastn.

Candidate gene allele specific qPCR

DNA was extracted from four bees from three Koppert colonies using the Qiagen DNA Micro kit according to manufacturer’s instructions. RNA was extracted from samples of the heads of the same worker bees with the QIAGEN RNeasy Mini Kit according to manufacturer’s instructions. cDNA was synthesized from a 8 µl sample of RNA using the Tetro cDNA synthesis Kit (Bioline, London, UK) as per manufacturer’s instructions.

We amplified numerous fragments of the 19 candidate genes. Sanger sequencing results were analyzed using the heterozygote analysis module in Geneious version 7.3.0 to identify heterozygotic nucleotide positions. It was difficult to identify snps in exonic regions of the 19 loci, which could be amplified with primers of suitable efficiency. We managed to identify a suitable region in toll-like receptor Tollo (AELG01000623.1 exonic region 1838–2420).

The locus was run for three different reactions; T allele, G allele and reference. Reference primers were designed according to Gineikiene, Stoskus & Griskevicius (2009). A common reverse primer (CTGGTTCCCGTCCAATCTAA) was used for all three reactions. A reference forward primer (CGTGTCCAGAATCGACAATG) was designed to the same target heterozygote sequence, upstream of the heterozygote nucleotide position. The reference primers measure the total expression of the gene, whereas the allele specific primers (T allele: CCAGAATCGACAATGACTCGT, G allele: CAGAATCGACAATGACTCGG) measure the amount of expression due to the allele. Thus the ratio between the allele specific expression and reference locus expression would be the relative expression due to the allele.

Three replicate samples were run for each reaction. All reactions were prepared by the Corbett robotics machine, in 96 well qPCR plates (Thermo Scientific, Loughborough, UK). The qPCR reaction mix (20 µl) was composed of 1 µl of diluted cDNA (50 ng/ µl), 1 µl of forward and reverse primer (5 µM/ µl each), 10 µl 2X SYBR Green JumpStart Taq ReadyMix (Sigma Aldrich, Irvine, UK) and 7 µl ddH20. Samples were run in a PTC-200 MJ thermocycler. The qPCR profile was; 4 min at 95 °C denaturation followed by 40 cycles of 30 s at 95 °C, 30 s at 59 °C and 30 s at 72 °C and a final extension of 5 min at 72 °C.

Forward primers are different, both in their terminal base (to match the snp) and in their length. It is entirely possible that they may amplify more or less efficiently even if there was no difference in amount of template (Pfaffl, 2001). To test for this we repeated all qPCRs with genomic DNA (1 µl of diluted DNA (20 ng/ µl) from the same bees as the template. We would expect equal amounts of each allele in the genomic DNA. We also measured efficiency of each reaction as per Liu & Saint (2002).

Median Ct was calculated for each set of three technical replicates. A measure of relative expression (ratio) was calculated for each allele in each worker bee as follows: (1) ratioallele=Eallele−CtalleleEreference−Ctreference

E is the median efficiency of each primer set (Liu & Saint, 2002; Pfaffl, 2001). All statistical analysis was carried out using R (3.3.1) (R core Team, 2016).

Results

Discovery of monoallelically methylated and expressed genes

In total, we found nineteen genes that were both monoallelically methylated (present in both Me-DIP and MRE-seq libraries) and monoallelically expressed (only one allele present in the RNA-seq library). Figures 1 and 2 show the coverage of the three libraries for two examples of these genes (ras GTPase-activating protein nGAP-like and bicaudal-D). Of the nineteen genes, fourteen had the hypermethylated (MeDIP) allele expressed, while five had the hypomethylated (MRE-seq) allele expressed (see Tables S1). The nineteen genes were compared to the nr/nt database using Blastn. Six returned noninformative hits (Table 2).

Figure 1 Coverage of the three libraries for ras GTPase-activating protein nGAP-like (LOC100652225).

The transcript models come from GCF_000214255.1_Bter_1.0. The y-axis in the coverage plots is log (1 + coverage). The red vertical line represents the heterozygote position. The MeDip allele was expressed in this locus, see Table 2.

Figure 2 Coverage of the three libraries for bicaudal D-related protein homolog ( LOC100650109).

The transcript model come from GCF_000214255.1_Bter_1.0. The y-axis in the coverage plots is log (1 + coverage). The red vertical line represents the heterozygote position. The MeDip allele was expressed in this locus, see Table 2.

Table 2 The thirteen of the nineteen monoallelically methylated and expressed genes that returned informative blast hits.

Gene	Accession	Expressed allele	Function	
yippee-like 1	LOC100642754	MeDIP	Yippee is an intracellular protein with a zinc-finger like domain. DNA methylation of a CpG island near the yippie-like 3 promoter in humans represents a possible epigenetic mechanism leading to decreased gene expression in tumours (Kelley et al., 2010).	
toll-like receptor Tollo	LOC100644648	MeDIP	Tollo regulates antimicrobial response in the insect respiratory epithelium (Akhouayri et al., 2011).	
zinc finger protein Elbow	LOC100650465	MeDIP	The elbow (elB) gene is involved in the formation of the insect tracheal system (Dorfman et al., 2002).	
heterogeneous nuclear ribonucleoprotein A3	LOC100651168	MeDIP	Heterogeneous nuclear ribonucleoproteins associated with precursors of functional, protein coding mRNAs (Dreyfuss et al., 1993).	
calmodulin-lysine N-methyltransferase-like	LOC100749522	MRE	Calmodulin-lysine N-methyltransferase catalyses the trimethylation of a lysine residue of calmodulin. Calmodulin is a ubiquitous, calcium-dependent, eukaryotic signalling protein with a large number of interactors. The methylation state of calmodulin causes phenotypic changes in growth and developmental processes (Magnani et al., 2010).	
Na/K/Ca exchanger CG1090	LOC107998466	MRE	CG1090 functions in the maintenance of calcium homeostasis.	
Shaker	LOC100648438	MeDIP	Shaker is involved in the operation of potassium ion channel. Shaker expression was upregulated in sterile versus reproductive honeybee workers (Cardoen et al., 2011).	
Centrosomal and chromosomal factor-like	LOC105665737	MeDIP	Essential protein required for proper condensation of mitotic chromosomes and progression through mitosis. Expressed during oogenesis in Drosophila (Kodjabachian et al., 1998).	
excitatory amino acid transporter 1	LOC100744217	MRE	Excitatory amino acid transporters are neurotransmitter transporters. Excitatory amino acid transporter 3 expression was upregulated in sterile honeybee workers (Cardoen et al., 2011). Excitatory amino acid transporter 1 expression differences were also associated with worker - queen differentiation in the paper wasp Polistes metricus (Toth et al., 2014).	
aminopeptidase M1-like	LOC105666993	MeDIP	M1 aminopeptidases are zinc-dependent enzymes that catalyze the removal of amino acids from the N terminus of polypeptides (Drinkwater et al., 2017).	
ras GTPase-activating protein nGAP-like	LOC100652225	MeDIP	Ras GTPase-activating protein 1 was found to be upregulated in reproductive honeybee workers (Cardoen et al., 2011). It is involved in oocyte meiosis.	
neuromedin-B receptor-like	LOC100745453	MeDIP	In humans, this G protein-coupled receptor binds neuromedin B, a peptide that stimulates mitosis in gastrointestinal epithelial tissue.	
bicaudal D-related protein homolog	LOC100650109	MeDIP	Bicaudal is involved in embryonic pattern formation in Drosophila (Markesich et al., 2000). It is thought to be involved in the differentiation between soldiers and workers in the termite Reticulitermes flavipes (Scharf et al., 2003). Bicaudal protein D has been shown to be methylated more in eggs than sperm in honeybees (Drewell et al., 2014).	

Confirmation of monoallelic expression

Monoallelic expression was confirmed in one of these nineteen (toll-like receptor Tollo (LOC100644648)) by allele specific qPCR (Amarasinghe et al., 2015). The allele with a guanine at the snp position had a mean expression of 6.04 ± 8.28 (standard deviation) in four bees from three different colonies. The thymine allele was not expressed at all in these bees. This was not due to the efficiency of the primers as the DNA controls of both alleles showed similar amplification (G mean =422.70 ± 507.36, T mean =1575.17 ± 503.02). In the three other loci tested (Ras GTPase-activating protein 1, LOC107964816, Elbow) we found apparent monoallelic expression, but could not dismiss primer efficiency as the cause.

We then looked at these nineteen genes in twenty-nine previously published RNA-seq libraries. Fifteen of these nineteen genes expressed a single allele in all twenty nine RNA-seq libraries, see Tables S2. The remaining four genes were inconsistent; they showed expression of one allele in some B. terrestris workers, and expression of two alleles in other workers.

Removing batch effects

The twenty nine RNA-seq libraries do not come from the same experiment (Table 1). This gives rise to the possibility of batch effects, sources of variation due to samples not being from the same source or not being run together. We must remove these before any other analysis.

The mean GC content of the 29 libraries was 42.34%, with individual libraries having a similar GC content ranging from 40–46%. GC content differed with run (Nested ANOVA: F = 20.302, df = 1, p < 0.001), but not by colony (Nested ANOVA: F = 1.763, df = 4, p = 0.171). The mean coverage of the 29 libraries was 13.29, with mean library coverage ranging from 9.84 to 17.61. Run had an effect on coverage (Nested ANOVA: F = 7.554, df = 1, p = 0.011), as did colony (Nested ANOVA: F = 6.962, df = 4, p < 0.001).

Therefore, the combat method in the R package SVA (version 3.20.0) was used to remove any batch effects and control for original differences in coverage (Leek et al., 2012; Johnson, Li & Rabinovic, 2007). The success of this control was confirmed by the R package edgeR (version 3.14.0) (McCarthy, Chen & Smyth, 2012; Robinson, McCarthy & Smyth, 2010). The SVA adjustment reduced the edgeR dispersion value from 3.9994 (BCV  = 2) to 0 (BCV  = 0.0003) (see Fig. 3). That is, we successfully removed the batch effects due to the separate runs.

Figure 3 Biological coefficient of variation (BCV) of (A) raw data, and (B) SVA-adjusted data for the 29 RNA-seq Bombus terrestris libraries.

The black dots represent the BCV if it were calculated individually for each gene (tagwise). The blue line is the trend of this data. The red line represents the BCV of the samples if a common dispersion value, over all genes, were used. In (B) tagwise values are exactly the same as common values so no black dots are visible.

Allele specific expression—RNA-seq

We then searched more generally for allele specific expression in the twenty-nine RNA-seq libraries. A total of 555 loci showed allele-specific expression in ≥3 of the 29 RNA-seq libraries (Tables S3). Comparing these loci against the Bombus terrestris genome using Blastn returned 211 hits. To search for gene ontology terms, we compared them against Drosophila melanogaster proteins, using Blastx, which returned 329 hits. We tested for enriched gene ontology (GO) terms against their background value in the bumblebee transcriptome. One hundred and fifty-one Gene Ontology(GO) terms were enriched in the 555 regions showing allele specific expression (Fisher’s exact test p > 0.05), however none were significant at the more stringent FDR > 0.05. Figure 4 shows the large number of biological functions associated with these 555 genes.

Figure 4 GO terms associated with allele specific expression.

A summary of the enriched GO terms (p < 0.05, based on Blast2Go annotation) found for genes displaying allele specific expression. This figure was produced using Revigo (Supek et al., 2011). Each rectangle represents a single cluster of closely related GO terms. These rectangles are joined into different coloured ‘superclusters’ of loosely related terms. The area of the rectangles represents the p-value associated with that cluster’s enrichment.

Discussion

An important caveat about the integrative analysis of monoallelic methylation and expression carried out here is that all three libraries were from a single bee. It is certain that there is variation in methylation and allele specific expression between bees just as there is in other species (Pignatta et al., 2014). We attempted to confirm this monoallelic expression in other bees using RNA-seq and qPCR but with limited success. This analysis is only a first step in understanding the link between monoallelic methylation and expression.

Of the nineteen genes displaying monoallelic methylation and monoallelic expression, fourteen had the hypermethylated (MeDIP) allele expressed, while five had the hypomethylated (MRE-seq) allele expressed (see Tables S1). In ant genes with allele specific methylation, the hypermethylated allele showed more expression than the hypomethylated allele (Bonasio et al., 2012). This fits with genome wide analysis that shows exonic methylation in insects associated with increased gene expression (Glastad, Hunt & Goodisman, 2014; Yan et al., 2015). Our fourteen genes with the hypermethylated allele expressed agree with this pattern. But how to explain the five genes where the hypomethylated allele was expressed? Firstly, the role of methylation in insect gene expression is not clear cut, with the relationship between exonic methylation and expression often disappearing at the gene level (Yan et al., 2015). For example, EGFR expression is lower in ant workers that exhibit higher DNA methylation of EGFR (Alvarado et al., 2015). Secondly, even in the canonical mammalian methylation system, the “wrong” allele has been shown to be expressed occasionally due to lineage specific effects (Dean et al., 1998; Pardo-Manuel de Villena, de la Casa-Esperón & Sapienza, 2000; Onyango et al., 2002; Sapienza, 2002; Zhang et al., 1993).

We analysed RNA-seq libraries from different published sources. This lead to two confounding problems. The first is that as the samples were run at different times, using different machines this could lead to a batch effect. We were able to successfully remove this. The second, that the libraries were made from abdomens in some cases and whole bodies in others, is still a confounding effect. Allele specific expression is known to vary between tissues (Chamberlain et al., 2015). Any variation in which allele is expressed could be due to these tissue effects.

We looked at the expression of the nineteen genes in all twenty-nine RNA-seq libraries. If they are monoallelically expressed in these bees, we would find only one allele in a given RNA-seq library. Fifteen of these nineteen genes were confirmed to show a single allele in all twenty-nine RNA-seq libraries. We would also find only one allele if that bee was homozygous. We cannot rule out that these fifteen genes just happen to be homozygous in all twenty-nine bees from five different colonies from multiple sources.

The remaining four genes showed inconsistent expression with one allele being expressed in some B. terrestris workers, and expression of two alleles in other workers. Natural intraspecific variation in allele specific expression has been found in other species (Pignatta et al., 2014). The tissue variation mentioned above is also a possibility. Another explanation is that these loci are not epigenetically controlled but rather their allele specific expression is derived from genetic effects (Remnant et al., 2016).

There are three main genetic, as opposed to epigenetic, affectors of allele specific expression (Edsgard et al., 2016). Allele specific expression can be caused by differences in the alleles’ sequence within the translated part resulting in a modified protein. A change at the alleles’ cis regulatory sites, could cause differential binding of transcription factors. Transcript processing can be affected by a change in the alleles’ sequence a splice site or untranslated region. This large number of possible causes of allele specific expression could explain why we see so many functions associated with the 555 genes showing allele specific expression (Fig. 4).

But it is not just allele specific expression that may have genetic as well as epigenetic effects. It has been shown in humans that some allele specific methylation is determined by DNA sequence in cis and therefore shows Mendelian inheritance patterns (Meaburn, Schalkwyk & Mill, 2010). An extreme example of genetically controlled allele specific methylation is found in Nasonia wasps, where there is no evidence for methylation driven allele specific expression but inheritable cis-mediated allele specific methylation has been found (Wang, Werren & Clark, 2016). This cis-mediated methylation has recently been suggested as being important in social insect biology (Remnant et al., 2016; Wedd, Kucharski & Maleszka, 2016).

We have found that allele specific expression is widespread in the bumblebee. We have also found that the extreme version of allele specific expression, monoalleic expression is associated with monoallelic methylation. Genomic imprinting in mammals usually involves monoallelic methylation and expression. Although tempting to associate our results with genomic imprinting, this current work is unable to identify genomic imprinting. In any case, caution should be applied due to the lack of understanding of the functional role of methylation in gene expression in insects and in the, as yet unquantified, role of genetic cis effects in insect allele specific methylation and expression.

Supplemental Information

Table S1 Nineteen genes showing both monoallelic methylation and monoallelic expression

Blast results and genomic coordinates of the reads from the RNA-seq, MRE-seq and MeDip-seq libraries.

Click here for additional data file.

Table S2 Confirmation of single allele expression of nineteen monoallelically expressed genes in twenty-nine previously published transcriptomes

For each of the 19 contigs, the previously published RNA-seq libraries with associated read counts are included.

Click here for additional data file.

Table S3 555 genes showing allele specific expression in at least three of the 29 previously published RNA-seq libraries

This table details the blast results from both the bumblebee and drosophila genomes and the GO terms associated with the drosophila hits.

Click here for additional data file.

Thanks to Sally Adams for helpful comments.

Additional Information and Declarations

Competing Interests

Author Contributions

Data Availability

The authors declare there are no competing interests.

Zoë Lonsdale, Kate Lee and Harindra Amarasinghe conceived and designed the experiments, performed the experiments, analyzed the data, wrote the paper, prepared figures and/or tables, reviewed drafts of the paper.

Maria Kiriakidu analyzed the data, wrote the paper, reviewed drafts of the paper.

Despina Nathanael conceived and designed the experiments, performed the experiments, wrote the paper, reviewed drafts of the paper.

Catherine J. O’Connor analyzed the data, prepared figures and/or tables.

Eamonn B. Mallon conceived and designed the experiments, wrote the paper, prepared figures and/or tables, reviewed drafts of the paper.

The following information was supplied regarding data availability:

All sequence data for this study are archived under NCBI BioProject numbers PRJEB9366 and PRJNA391408. GO-analysis results and lists of differentially expressed transcripts are available as Supplemental Information.

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
