# Peer review of "Allele specific expression and methylation in the bumblebee, Bombus terrestris"

_PeerJ, doi:10.7717/peerj.3798_

## Round 0.1 · original submission · Major Revisions

Please see the reviewer comments. Both are quite positive, but also suggest some critical issues, particularly with regard to the figures and suggest some additional data that could be presented.

I encourage you to address these comments to the best of your ability and look forward to a resubmission.

Reviewer 1 ·

Basic reporting

The manuscript is well written with clear and professional english used throughout. Literature is appropriately referenced and sufficient context to the paper is given.

The figures are quite disappointing and do not add to the manuscript. In the main manuscript, only one figure is shown (gene ontology) and this is not discussed adequately in the text. What is the biological significance of these enrichment categories? What conclusions do the authors derive from this? The supplemental figure is not well described and confusing. Should there be dots in figure b? What does 'tagwise' mean?

I would really like to see a gene browser view of one/some of the genes identified in this study, showing the distribution of reads from the three sequencing methods around the gene. This would really help give a sense of the data to the reader and help visualise the results and concepts trying to be conveyed. Much more data can be presented in the figures which currently are very limited.

Table 2 is missing.

line 188 has incomplete information: "x, x"

Experimental design

The manuscript is within the scope of PeerJ. The research question is well defined, relevant and meaningful and worth investigating.

The methods are generally well described with the exception of the bioinformatic analysis of the methylation data which could benefit from additional explanation. Additionally, regarding the gene ontology analysis, could the authors clarify what background gene set was used to perform the analysis.

I am unclear whether the combination of MRE-seq and MeDip is suitable to identify allele-specific methylation given the inherent biases given by each method. Could the authors provide supporting information or validated bioinformatic methods to confirm that the results seen are not an artefact seen by the different enrichment methods? Alternatively, would it be possible to validate one or more of the loci by bisulfite-PCR based methods?

Validity of the findings

While the hypothesis is interesting and deserves investigation, the study does not appear to be developed but remains at a preliminary and unfinished stage. Unfortunately the manuscript and figures reflect this.

There is clear speculation regarding the role of genomic imprinting in this context. While this has been identified as speculative, there is absolutely no evidence suggesting that this is the case. Of the validated expression patterns, none showed a clear bias of one allele in one individual and the reciprocal allele in another which would be supportive (but not conclusive) of imprinting. It is more likely that the results are due to underlying genomic variations which are affecting the expression of the allele in cis. I would suggest to tone down the claims to genomic imprinting, especially in the abstract.

Additional comments

Great hypothesis and I was really caught and intrigued by the end of the introduction. Unfortunately I find that the results are minimal and left me wanting a fuller and deeper description of what is happening in this very interesting system.

Reviewer 2 ·

Basic reporting

The article was generally well written. The ideas were clear. The questions were put into appropriate scientific context.

Experimental design

See 'General Comments'.

Validity of the findings

See 'General Comments'.

Additional comments

Review of “Allele specific expression and methylation in the bumblebee, Bombus terrestris” by Lonsdale et al, PeerJ ms #18128.

The authors investigate the relationship between allele-specific expression (ASE) and DNA methylation in the bumblebee. They find some evidence of a link between ASE and gene methylation. They interpret their results in light of the importance of imprinting and control of gene regulation in insects.

The study of ASE is of considerable importance to the scientific community. The subject is of particular interest to those studying social systems, because of the possible importance of gene imprinting in these cases. Thus I think the general subject area is worthy of study.

The authors did find some evidence for a relationship between methylation and ASE in their samples. However, I was somewhat skeptical of how strong this finding was, given the methods used. That is, bisulfite-sequencing combined with RNA-Seq is the gold standard for studying methylation-expression relationships. But the methods used here, and the use of data from a variety of studies, were not as strong. Nevertheless, I still felt the study was worthy given the context of the study and journal. But I would like to see the authors add a few more caveats to their Discussion regarding possible issues with their analyses.

Comments

The Introduction is well written and reflects the state of the field. I particularly like the fact that the authors accurately discuss the lack of clarity and consensus surrounding the field of epigenetics in social insects.

82 It looks like the authors have only one ‘replicate’ of their laboratory work (i.e., they only analyzed one bee). This is unfortunate as replication of methylation analyses is now the standard of work. I am sympathetic to the difficulty in being able to carry out these studies. But the use of only one replicate weakens the study and thus the results here must be considered with some caution.

90 I’m not familiar with MRE-Seq and was a little confused about what was being derived from this analysis. The authors state that “MRE-seq enriches for unmethylated cytosines” and so I expected that this would mean sequences obtained from the MRE-Seq analyses would represent unmethylated sections of DNA. But later (106) the authors state that “We searched for genes that were monoallelically methylated (present in both MeDip-seq and MRE-seq libraries)” which suggests that MRE-seq procedure is actually enriching for methylated, rather than unmethylated, DNA. Can the authors be clearer about this?

106 Following up on the query above, the authors state that “We searched for genes that were monoallelically methylated (present in both MeDip-seq and MRE-seq libraries)”. But does this mean that they discovered one allele in the MeDip-seq library (the putatively ‘methylated’ allele) and the other allele in the MRE-seq library (the putatively unmethylated allele)? Is that how they identified monoallelically methylated alleles? Please be clear.

It seems that at least some of the RNA-Seq analyses were not conducted on the same bees (or even the same tissues?) as was used for the MeDIP and MRE analyses. Is this correct? If so, this should be acknowledged and discussed. It is possible that the link between methylation and ASE may be tissue specific which might confound the authors' analyses.

In general, the authors should discuss from which tissues (brains? Whole bodies?) the various samples were obtained. This could be quite important if they are trying to link methylation to expression. Did all studies analyze the same tissues? If not, this needs to be discussed.

Detecting heterozygosity isn’t trivial. For example, a single sequencing error might give the appearance of heterozygosity in the sample even if it isn’t real. In principle, there are ways of handling this. You might expect ‘true’ heterozygotes to show 50% of each sequencing allele. The authors should discuss this a bit and explain how they differentiated true heterozygotes for artifacts.

132 The authors provide helpful information on the ‘coverage’ of their libraries. This is certainly useful. But it does raise the question of how allele specific expression was detected and how accurate it is. This is not a trivial issue, as the detection and confidence that an allele is showing allele-specific expression depends on the number of reads that come from that allele, which may depend on the length of the gene and other factors. These issues have been considered before and they deserve a bit more consideration and discussion here.

194 Much of the analyses the authors undertake relies on the counts of sequences obtained in their libraries. The authors presumably excluded genes that had low count numbers (e.g., a gene have only a single count in the RNA-Seq library means that it would always be viewed as showing ASE). But knowing the count numbers for each gene in each of their libraries would provide insight into the robustness of the results. Could this information be provided in Sup Table 1?

I was little confused as to whether figure 1 had some kind of meaning I wasn’t getting. Is this essentially just a list of GO terms associated with genes displaying allele specific expression? Or does the structure of the figure have any meaning? For example, does the fact that certain terms are at the top of the box have any meaning? Or does the fact that some terms are next to each other (e.g., hatching behavior and phosphate-containing compound) have any meaning beyond what is conveyed by the colors. If this figure actually does not convey any meaning (as a figure) the information may better be provided in a table.

The Discussion is generally well written and acknowledges the complexities of interpretation of methylation information in insects.

---

## Round 0.2 · accepted · Accept

Good news to start off your week! I'm happy to pass along a decision to Accept your manuscript for publication in PeerJ. Congratulations!

Reviewer 1 ·

Basic reporting

This is a revised manuscript by Lonsdale et al. investigated monoallelic DNA methylation and expression in the bumblebee. Overall I find the manuscript much improved and my initial concerns have now been satisfied.

The language, article structure and discussion all read well and the manuscript is well written and easy to read. Appropriate citations are used. The discussion of genomic imprinting has now been toned down as appropriate.

Experimental design

The additional figures and explanations in the text greatly add to the manuscript and increase the clarity of what was done and what conclusions can be made from the data.

Validity of the findings

Speculative results have now been put in context and caveats are now listed as appropriate.

Additional comments

A very nicely revised manuscript which I agree is a much improved version of the original submission. I look forward to seeing it in print!

---

## Author Rebuttal · Round 0.2

Social Epigenetics Lab
Department of Genetics and Genome Biology
University of Leicester, UK

University of Arizona

August 11th, 2017

Dear Prof. Berghout,

Thank you and the two reviewers for your time and insight in reviewing our manuscript. We have made the alterations suggested. These are detailed below with reviewers comments numbered and italicised and our response in normal font. Line numbers refer to the lines in the latest version of the manuscript. We feel, and I hope you and the reviewers agree, this is a much improved manuscript.

## Editor's comments

1. *"Both are quite positive, but also suggest some critical issues, particularly with regard to the figures and suggest some additional data that could be presented."*

   - We have added three new figures to the main text.
   - We have more clearly explained the meaning of the treemap figure (now Figure 4) and BCV figure (now Figure 3).
   - We have added tissue information to table 1.
   - Table 2 should now be visible.
   - We now supply the SRA data required.
   - We now provide the background gene set used for the enrichment analysis.
   - We provide background information on the validity of the integrative monoallelic methylation and expression analysis.
   - We explained the logic of the QuaSAR analysis.
   - We demphasize the possible role of genomic imprinting in our discussion.

**Reviewer 1**

1. *The figures are quite disappointing and do not add to the manuscript. In the main manuscript, only one figure is shown (gene ontology) and this is not discussed adequately in the text. What is the biological significance of these enrichment categories? What conclusions do the authors derive from this? The supplemental figure is not well described and confusing. Should there be dots in figure b? What does 'tagwise' mean?*
There are three points here.
1) There is a lack of useful figures. We have included two new figures (see answer to Reviewer 1 comment 2), moved the BCV figure from the supplemental to Figure 3. In total there are now four figures.
2) Gene ontology figure is poorly described. Please see our response to Reviewer 2 comment 9.
3) Supplemental figure is not well described. This figure is now in the text as Figure 3. We have have added a substantial amount of explanation to it.
Lines 226 - 240 (new parts in bold) now read:
"**The twenty nine RNA-seq libraries do not come from the same experiment (Table 2). This gives rise to the possibility of batch effects, sources of variation due to samples not being from the same source or not being run together. We must remove these before any other analysis.**
The mean GC content of the 29 libraries was 42.34%, with individual libraries having a similar GC content ranging from 40-46%. GC content differed with run (Nested ANOVA: F = 20.302, df = 1, p < 0.001), but not by colony (Nested ANOVA: F = 1.763, df = 4, p = 0.171). The mean coverage of the 29 libraries was 13.29, with mean library coverage ranging from 9.84 to 17.61. Run had an effect on coverage (Nested ANOVA: F = 7.554, df = 1, p = 0.011), as did colony (Nested ANOVA: F = 6.962, df = 4, p < 0.001).
**Therefore, the combat method in the R package SVA (version 3.20.0) was used to remove any batch effects and control for original differences in coverage (Leek et al., 2012; Johnson et al., 2007). The success of this control was confirmed by the R package edgeR (version 3.14.0) (McCarthy et al., 2012; 227 Robinson et al., 2010). The SVA adjustment reduced the edgeR dispersion value from 3.9994 (BCV=2) to 0 (BCV=0.0003) (see Figure 3). That is we successfully removed the batch effects due to the separate runs.**"
The Legend for Figure 3 now reads:
"Biological coefficient of variation (BCV) of a) raw data, and b) SVA-adjusted data for the 29 RNA-seq *Bombus terrestris* libraries. **The black dots represent the BCV if it were calculated individually for each gene (tagwise). The blue line is the trend of this data. The red line represents the BCV of the samples if a common dispersion value, over all genes, were used. In (b) tagwise values are exactly the same as common values so no black dots are visible.**"

2. *I would really like to see a gene browser view of one/some of the genes identified in this study, showing the distribution of reads from the three sequencing methods around the gene. This would really help give a sense of the data to the reader and help visualise the results and concepts trying to be conveyed. Much more data can be presented in the figures which currently are very limited.*
We have included two new gene browser-like figures (Figure 1 (*ras GTPase-activating protein nGAP-like*) and Figure 2 (*bicaudal D-related protein homolog*)). We agree with

the reviewer that these add value.

3. *Table 2 is missing.*
   Table 2 is the sideways table currently on page 8. In the original pdf it was on page 6. We apologise that this was not viewable to the reviewer.

4. *line 188 has incomplete information: "x, x"*
   At the time of the original submission, we had not finished the SRA upload. Lines 196 -197 now reads:
   **"All sequence data for this study are archived at the NCBI Sequence Read Archive (SRA) Accession no. PRJEB9366 and PRJNA391408."**

5. *The methods are generally well described with the exception of the bioinformatic analysis of the methylation data which could benefit from additional explanation.*
   We have added a substantial amount of explanation in response to other comments from reviewers. Please see the responses to reviewer 1 comment 7, reviewer 2 comment 1, reviewer 2 comment 2.

6. *Additionally, regarding the gene ontology analysis, could the authors clarify what background gene set was used to perform the analysis.*
   We have added
   Line 154-157: **"We carried out an enrichment analysis (Fisher exact test) using a custom R script (https://dx.doi.org/10.6084/m9.figshare.3201355.v1) on this list of GO terms. This identified GO terms that are over-represented (p <0.05) relative to the entire bumblebee transcriptome (https://dx.doi.org/10.6084/m9.figshare.3458828.v1)."**
   Line 246: **"We tested for enriched gene ontology (GO) terms against their background value in the bumblebee transcriptome."**

7. *I am unclear whether the combination of MRE-seq and MeDip is suitable to identify allele-specific methylation given the inherent biases given by each method. Could the authors provide supporting information or validated bioinformatic methods to confirm that the results seen are not an artefact seen by the different enrichment methods? Alternatively, would it be possible to validate one or more of the loci by bisulfite-PCR based methods?*
   We based this assay on the Harris 2010 paper, which establishes this technique in human work. They independently validated it. We have added the following to lines 54 - 58:
   **"In this paper, we examined the link between monoallelic methylation and monoallelic expression in the bumblebee, *Bombus terrestris* using an integrative approached previously used in human epigenetic studies (Harris et al 2010). Namely, we compare two types of whole methylome libraries and an RNA-seq library from the same individual. In humans, this integrative approach has been independently validated by clonal bisulphite sequencing (Harris et al 2010)."**

8. *There is clear speculation regarding the role of genomic imprinting in this context. While this has been identified as speculative, there is absolutely no evidence suggesting that this is the case. Of the validated expression patterns, none showed a clear bias of one allele in one individual and the reciprocal allele in another which would be supportive (but not conclusive) of imprinting. It is more likely that the results are due to underlying genomic variations which are affecting the expression of the allele in cis. I would suggest to tone down the claims to genomic imprinting, especially in the*

*abstract.*

We have toned down the discussion of genomic imprinting. The abstract now has no mention of genomic imprinting, the relevant section now reads:

**"We discuss our results with reference to the functional role of methylation in gene expression in insects and in the, as yet unquantified, role of genetic cis effects in insect allele specific methylation and expression."**

We have altered Lines 304 -307 to make clear our current work is unable to identify genomic imprinting and to put the emphasis on cis effects.:

**"Although tempting to associate our results with genomic imprinting, this current work is unable to identify genomic imprinting. In any case, caution should be applied due to the lack of understanding of the functional role of methylation in gene expression in insects and in the, as yet unquantified, role of genetic cis effects in insect allele specific methylation and expression"**.

## Reviewer 2

1. *It looks like the authors have only one replicate of their laboratory work (i.e., they only analyzed one bee). This is unfortunate as replication of methylation analyses is now the standard of work. I am sympathetic to the difficulty in being able to carry out these studies. But the use of only one replicate weakens the study and thus the results here must be considered with some caution.*

   We agree with the reviewer and have added the following to the beginning of the Discussion (lines 252 -257):

   **"An important caveat about the integrative analysis of monoallelic methylation and expression carried out here is that all three libraries were from a single bee. It is certain that there is variation in methylation and allele specific expression between bees just as there is in other species (Pignatta et al. 2014). We attempted to confirm this monoallelic expression in other bees using RNA-seq and qPCR but with limited success. This analysis is only a first step in understanding the link between monoallelic methylation and expression."**

2. *Im not familiar with MRE-Seq and was a little confused about what was being derived from this analysis. The authors state that MRE-seq enriches for unmethylated cytosines and so I expected that this would mean sequences obtained from the MRE-Seq analyses would represent unmethylated sections of DNA. But later (106) the authors state that We searched for genes that were monoallelically methylated (present in both MeDip-seq and MRE-seq libraries) which suggests that MRE-seq procedure is actually enriching for methylated, rather than unmethylated, DNA. Can the authors be clearer about this? Following up on the query above, the authors state that We searched for genes that were monoallelically methylated (present in both MeDip-seq and MRE-seq libraries). But does this mean that they discovered one allele in the MeDip-seq library (the putatively methylated allele) and the other allele in the MRE-seq library (the putatively unmethylated allele)? Is that how they identified monoallelically methylated alleles? Please be clear.*

   Yes that is exactly how we identified monoallelically methylated loci. We have altered several parts of the text to make this clearer.

   Lines 61-63:

   **"Genes found in both libraries are predicted to be monoallelically methylated, with the putatively hypermethylated allele being in the MeDIP-seq**

**data and the putatively hypomethylated allele in the MRE-seq data**"
Lines 108 -111
"**We searched for genes that were monoallelically methylated (present in both MeDip-seq (the putatively hypermethylated allele) and MRE-seq (the putatively hypomethylated allele) libraries), heterozygous (different alleles in the methylation libraries) and monoallelically expressed (only one allele present in the RNA-seq library).**"

3. *It seems that at least some of the RNA-Seq analyses were not conducted on the same bees (or even the same tissues?) as was used for the MeDIP and MRE analyses. Is this correct? If so, this should be acknowledged and discussed. It is possible that the link between methylation and ASE may be tissue specific which might confound the authors' analyses. . . . In general, the authors should discuss from which tissues (brains? Whole bodies?) the various samples were obtained. This could be quite important if they are trying to link methylation to expression. Did all studies analyze the same tissues? If not, this needs to be discussed.*

The reviewer is correct that different colonies were used. This is noted in Table 1 and in the methods. The MRE/MeDIP/RNA-seq analysis, the 29 published libraries analysis and the qPCR are separate analyses so it is not important that they are the same bees. But within the 29 published RNA-seq analysis, this source of variation is of vital importance. That is why we carried out the batch effect analysis to reduce the variation due to this (see our answer to Reviewer 1 comment 1 for more detailed discussion of the batch effect).

The tissues used are varied. We describe them in the methods. To make it clear we have added them to Table 1. The reviewer is correct that we should discuss this source of variation. We have added the following (lines 271-276):

"**We analysed RNA-seq libraries from different published sources. This lead to two confounding problems. The first is that as the samples were run at different times, using different machines this could lead to a batch effect. We were able to successfully remove this. The second, that the libraries were made from abdomens in some cases and whole bodies in others, is still a confounding effect. Allele specific expression is known to vary between tissues (Chamberlain et al. 2015). Any variation in which allele is expressed could be due to these tissue effects.**"
Lines 284-285 now read:
"**The tissue variation mentioned above is also a possibility.**"

4. *Detecting heterozygosity isnt trivial. For example, a single sequencing error might give the appearance of heterozygosity in the sample even if it isnt real. In principle, there are ways of handling this. You might expect true heterozygotes to show 50% of each sequencing allele. The authors should discuss this a bit and explain how they differentiated true heterozygotes for artifacts . . . authors provide helpful information on the coverage of their libraries. This is certainly useful. But it does raise the question of allele specific expression was detected and how accurate it is. This is not a trivial issue, as the detection and confidence that an allele is showing allele-specific expression depends on the number of reads that come from that allele, which may depend on the length of the gene and other factors. These issues have been considered before and they deserve a bit more consideration and discussion here.*

We have combined the two points above as they relate to our QuaSAR analysis. The reviewer is asking 1) how did we detect heterozygousity, and 2) how did we detect

allele specific expression. QuaSAR does both steps (genotyping = detecting heterozygousity, inference of ASE = detecting allele specific expression). From the published QuaSAR paper "QuASAR, quantitative allele-specific analysis of reads, a novel statistical learning method for jointly detecting heterozygous genotypes and inferring ASE. The proposed ASE inference step takes into consideration the uncertainty in the genotype calls, while including parameters that model base-call errors in sequencing and allelic over-dispersion. We validated our method with experimental data for which high-quality genotypes are available. Results for an additional dataset with multiple replicates at different sequencing depths demonstrate that QuASAR is a powerful tool for ASE analysis when genotypes are not available." We have added a new paragraph (lines 140 - 150):

**"The R package, QuASAR implements a statistical method for: 1) genotyping from next-generation sequencing reads (according to the Hardy-Weinberg equilibrium), and 2) conducting inference on allele specific expression at heterozygous sites (Harvey et al. 2015). One problem with genotyping heterozygotes is being able to identical true homozygotes that appear heterozygote due to base-calling errors. QuASAR removes snps with extreme differential allele expression from the analyses, thus controlling for any base-calling errors. Despite this inherent conservatism, in benchmark tests, QuaSAR can accurately genotype loci with lower error rates than other methods commonly used for genotyping DNA-seq data (Harvey et al. 2015). The allele specific expression inference step takes into consideration the uncertainty in the genotype calls, base-call errors in sequencing, and allelic over-dispersion. QuASAR is a powerful tool for detecting allele specific expression if, as during most RNA-seq experiments, genotypes are not available (Harvey et al. 2015)."**

5. *Much of the analyses the authors undertake relies on the counts of sequences obtained in their libraries. The authors presumably excluded genes that had low count numbers (e.g., a gene have only a single count in the RNA-Seq library means that it would always be viewed as showing ASE). But knowing the count numbers for each gene in each of their libraries would provide insight into the robustness of the results. Could this information be provided in Sup Table 1?*

We say in lines 122 -124: "Only regions with a read depth of at least six in each of the libraries (RNA-seq, MeDIP-seq and MRE-seq) was used.". The read depths are now included in supplemental table 1. The new Figures 1 and 2 give two graphical examples of coverage of each of the libraries over two genes.

6. *I was little confused as to whether figure 1 had some kind of meaning I wasnt getting. Is this essentially just a list of GO terms associated with genes displaying allele specific expression? Or does the structure of the figure have any meaning? For example, does the fact that certain terms are at the top of the box have any meaning? Or does the fact that some terms are next to each other (e.g., hatching behavior and phosphate-containing compound) have any meaning beyond what is conveyed by the colors. If this figure actually does not convey any meaning (as a figure) the information may better be provided in a table.*

We find Revigo's treemaps a useful way of summarizing long list of GO terms. All similar GO terms are clustered into a single rectangles. The area of this rectangle represents the enrichment p value. Loosely related rectangles are then coloured the same to help the viewer make sense of this still long list. We have altered the text to

make this clearer. Lines 158-159 now read:

"**REVIGO summarizes lists of GO terms using a clustering algorithm based on semantic similarity measures.**"

The legend for figure 4 now reads:

"GO terms associated with allele specific expression. A summary of the enriched GO terms (p <0.05, based on Blast2Go annotation) found for genes displaying allele specific expression. This figure was produced using Revigo **(Supek et al. 2011). Each rectangle represents a single cluster of closely related GO terms. These rectangles are joined into different coloured superclusters of loosely related terms. The area of the rectangles represents the p-value associated with that cluster's enrichment**".

Sincerely yours,

Eamonn Mallon